# Time-clustering of wave storms in the Mediterranean Sea

Giovanni Besio[1], Riccardo Briganti[2], Alessandro Romano[3], Lorenzo Mentaschi[4], and Paolo De Girolamo[3]

[1]Department of Civil, Chemical and Environmental Engineering, University of Genoa, Italy
[2]Department of Civil Engineering, University of Nottingham, UK
[3]Department of Civil, Architectural and Environmental Engineering, La Sapienza University, Rome, Italy
[4]European Commission, Joint Research Centre (JRC), Ispra, Italy

*Correspondence to:* Giovanni Besio (giovanni.besio@unige.it)

**Abstract.** In this contribution we identify storm time-clustering in the Mediterranean Sea through **a comprehensive analysis of the the Allan Factor**. This parameter is evaluated from long time series of wave height provided by oceanographic buoy measurements and hindcast re-analysis of the whole basin, spanning the period 1979-2014 and characterized by a horizontal resolution of about 0.1 degree in longitude and latitude and a temporal sampling of one hour (Mentaschi et al., 2015). **The nature of the processes highlighted by the AF and the spatial distribution of the parameter are both investigated. Results reveal that the Allan Factor follows different curves at two distinct time scales. The range of time scales between 12 hrs to 50 days is characterised by a departure from the Poisson distribution. The second range, associated to timescales above 50 days, reveals seasonal fluctuations.** Transitional regimes are present at some locations in the basin. The spatial distribution of the Allan Factor reveals that the clustering at smaller time scales is present in the North-West of the Mediterranean, while **seasonality** is observed in the whole basin. This analysis is believed to be important to assess the local increased flood and coastal erosion risks due to storm clustering.

## 1 Introduction

In recent years the occurrence of different coastal storms in a short time has been studied in the context of storm driven erosion of beaches and dunes. Indeed it has been showed by different authors (Vousdoukas et al., 2012, Coco et al., 2014, Splinter et al., 2014; Karunarathna et al., 2014; Dissanayake et al., 2015) that storms occurring in quick succession may result in greater beach erosion than the cumulated erosion induced by single storms of far higher return periods.

In the events analysed in the aforementioned studies both the surge and the wave components played an important role. While studies that identify time-clustering of storm surges are available (e.g. Wadey et al., 2014, Haigh et al., 2016), there is no study, to the best knowledge of the authors, that analyses the clustering properties of wave storms alone. In micro-tidal environments, such as the Mediterranean Sea, wave storms are the principal driver of short term coastal erosion and flooding, hence it is important to understand the occurrence of clustering. The Mediterranean Sea wave climate has been extensively studied (e.g. Sartini et al., 2015a) and it is known that throughout the basin winter is richer in cyclones and, in turn, in wave storms. However, regional differences are significant. Sartini et al. (2015a) linked the seasonality of wave storms to seasonal features

of atmospheric pressure over the Mediterranean basin strongly suggesting that the local typical meteorological conditions determine different temporal regimes of storm waves.

The present work addresses the gap in the knowledge of the occurrence of time-clustering of wave storms by carrying out an analysis of wave storms sequences using the Allan Factor (hereinafter AF, Allan, 1966; Barnes and Allan, 1966), a well established technique to study the time behaviour of environmental processes. **When the underlying process is characterised by clustering, the AF of a specific sequence of events is larger than $1$ and shows a power-law behaviour at the time scales the exhibit departure from a Poisson distribution.** The simplicity of the AF analysis made it popular in the study of time sequences of a number of physical processes such as earthquakes (Telesca et al., 2002, Cavers and Vasudevan, 2015), lightning (Telesca et al., 2008), rainfall (Telesca et al., 2007, García-Marín et al., 2008) or fires (Telesca and Pereira, 2010). **However, the AF can be larger than $1$ and it can show a power-law behaviour also for non-homogeneous Poisson processes, as shown in Serinaldi and Kilsby (2013). Serinaldi and Kilsby (2013) and Telesca et al. (2012) provide methodologies to identify the nature of the process involved. Here we analyse the AF on long time series of wave height in the Mediterranean Sea provided by hindcast re-analysis spanning the period 1979-2014 (Mentaschi et al., 2015) and we apply the methodology proposed in Serinaldi and Kilsby (2013) to gain an insight in the type of process that is described by the AF**. This analysis is validated and compared against the AF evaluated using the time series of wave measurements of the Italian national Sea Wave Measurement Network (Rete Ondametrica Nazionale, hereinafter RON). The objective of this study is to identify the presence of time-clustering of wave storms in the whole Mediterranean basin and examine the time scales at which events are correlated as well as the spatial distribution of the clustering. To this end we identify scaling properties of wave storms in the wave hindcast database and map them over the Mediterranean Sea. The paper is organised as follows: after this Introduction, Section 2 explains the methodology used for the AF analysis, Section 3 describes the datasets used, Section 4 illustrates the results and Section 5 discusses the results and draws the conclusions of this work.

## 2 Clustering analysis methodology

Sequences of natural events such as earthquakes, rainfall, wildfires, can be seen as realisations of stochastic point processes. A process of this kind describes events that occur randomly in time and it is completely defined by the times at which these events occur. Here time series of sea states are considered. Each sea state is defined by a set of spectral parameter, such as the significant wave height $H_s$, the peak period $T_p$, the mean period $T_{m-1,0}$ and the mean direction of propagation $\theta_m$. Waves are always present on the sea surface, hence a sequence of storms need to be extracted from a time series of sea states by considering only events that satisfy a certain criterion. A storm is commonly defined as a sequence of sea states in which $H_s$ exceeds a given threshold (e.g. Goda, 1988). In this work, a threshold for each node is defined by considering the local 98% percentile of the $H_s$ distribution, regardless of $\theta_m$ (omnidirectional analysis, see figure 1 for threshold values of $H_s$ obtained with the hindcast model used here). The time $t_i$ at which the threshold is exceeded for the first time in each storm defines the event as part of a point process. If the interval between two subsequent events is below 12 hours, the two are regarded as one event, this is common practice in analysing storms and the value is deemed appropriate for the Mediterranean Sea

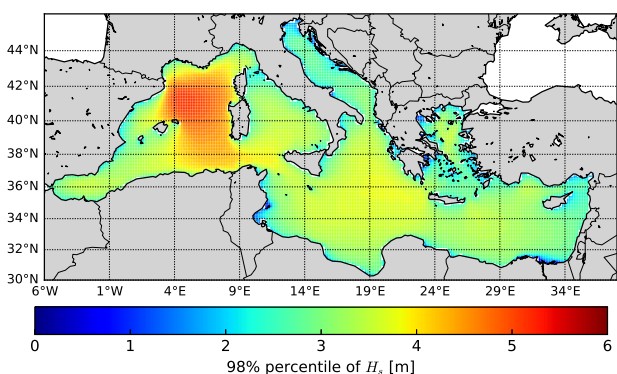

**Figure 1.** Value of significant wave height threshold in meters for the 98% percentile

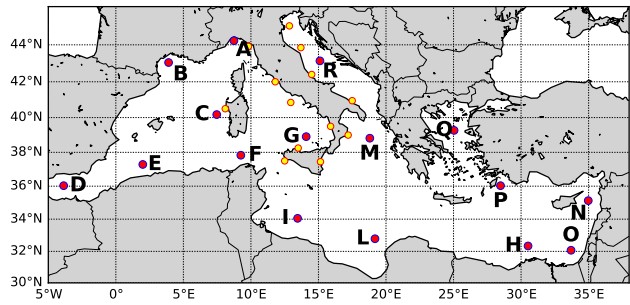

**Figure 2.** Hindcast control grid points (red circle) and RON buoys reference points (yellow circles)

(e.g. Sartini et al., 2015a). Therefore, in each of the computational nodes over the Mediterranean Sea (see figure 2 for a map of the domain and the location of few control grid points used in this study to show the single point behavior of the AF), a point process is defined. An example for the control point A and for the years 2004 and 2005, is given in figure 3. In this figure it is evident that most of the storms, during the two years considered, occur between November and May, showing the

5   pronounced seasonality that characterizes the basin. Figure 4 shows the number of events defined in each month over the year in the hindcast record for the same reference point A during the period 1979-2014 as a function of the percentile threshold (different wave heights). The seasonal variability of the storms in the Mediterranean basin is again recognizable. Note that the difference in number of storms between the different percentiles considered is maximum in the most active months and, if the 99% is chosen, the differences among seasons are small, although the seasonal variability is still recognizable.

10   Each of these processes is studied by defining equally spaced time windows of duration $\tau$ and counting the events in each window. The result is a sequence of counts $N_k$ ($k = 1, .., M$, where $M$ is the number of time windows). The clustering of the events is then studied with the Allan Factor (Allan, 1966; Barnes and Allan, 1966), defined as the variance of successive counts

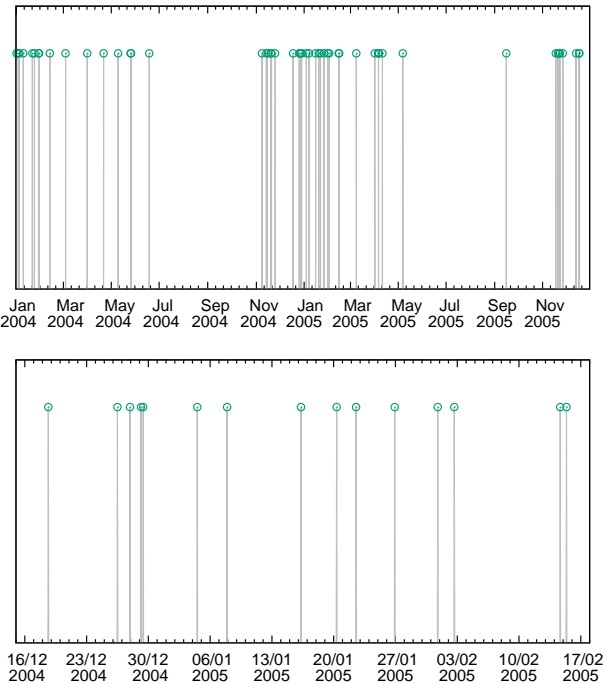

**Figure 3.** Storm occurence for the Northern Thyrrenian reference point (A): 2004/2005, top panel; zoom on winter 2004/2005, bottom panel

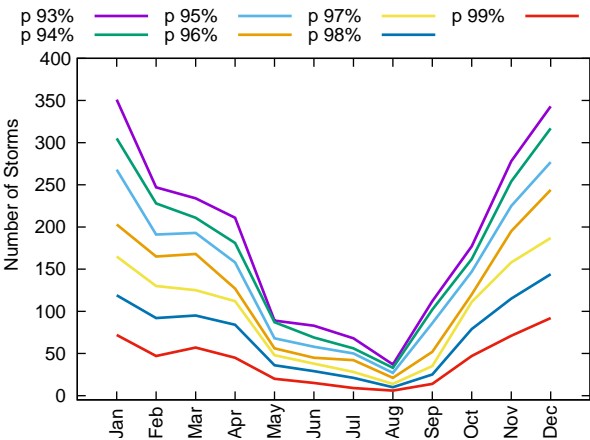

**Figure 4.** Number of Storms vs Threshold for the Northern Thyrrenian reference point (A)

as:

$$\mathrm{AF}\left(\tau\right) = \frac{\left\langle \left[N_{k+1}\left(\tau\right) - N_k\left(\tau\right)\right]^2 \right\rangle}{2\left\langle N_k\left(\tau\right)\right\rangle} \tag{1}$$

In general term, a point process is called fractal when a number of the relevant statistics shows scaling with related scaling exponents (Lowen and Teich, 1995). This implies that the AF depends on $\tau$ with a power-law, with exponent $\alpha$, which indicates the presence of clusters of points over a number of time scales $\tau$. For a fractal process with $0 < \alpha < 3$ this power law reads (Telesca and Pereira, 2010):

$$\mathrm{AF}(\tau) = 1 + \left(\frac{\tau}{\tau_1}\right)^{\alpha} \tag{2}$$

where $\tau_1$ is the fractal onset time that marks the lower limit for significant scaling behavior for the AF. For times smaller than $\tau_1$ there is no significant time correlation, while for times greater than $\tau_1$ a characteristic fractal trend can be derived from the value of the exponent. If the storms process is Poissonian, the arrival times are uncorrelated, hence $\alpha$ is expected to be zero and the AF will be near unity. If non-poissonian processes are present over a significant range of time scales it will be possible

to identify $\alpha > 0$ and AF$> 1$. In the present work $\alpha$ is estimated by the slope of the curve of the AF as a function of $\tau$. Note different ranges of $\tau$ can reveal different time scaling (clustering) of the same process through different slopes of eq. (2) due to different kind of forcing (Telesca and Pereira, 2010).

**Serinaldi and Kilsby (2013) demonstrated that cyclic, hence non-homogenous, Poisson processes show $AF > 1$ and power law behaviour for time scales associated to cyclic components. It is therefore necessary to compare the AF pattern**

**found in the wave time series with that of a process of known properties. A cyclic Poisson process is generated here with the same "integrate and fire" (IF) technique used in Serinaldi and Kilsby (2013). The cyclic components are selected by looking at the dominant harmonic components obtained with the Fourier analysis.**

## 3 Wave data

### 3.1 Wave hindcast

Wave hindcast in the Mediterranean Sea has been implemented on a time window covering 36 years, from the first of January of 1979 till the $31^{st}$ of December of 2014 (www.dicca.unige.it/meteocean/hindcast.html). The wave model is forced by the 10-m wind fields obtained by means of the non-hydrostatic model WRF-ARW (Weather Research and Forecasting - Advanced Research WRF) version 3.3.1 (Skamarock et al., 2008). In the present study a Lambert conformal grid covering the whole Mediterranean Sea with a resolution of about 0.1 degree in longitude and latitude has been used. Initial and boundary conditions

for atmospheric simulations were provided from the CFSR (Climate Forecast System Reanalysis) database (Saha et al., 2010). Use of CFSR reanalysis data for wave modeling provides reliable results, even if sometimes extreme wave conditions are not properly modeled (Cavaleri, 2009; Cox et al., 2011; Splinder et al., 2011; Carvalho et al., 2012; Chawla et al., 2013). For further details of the set-up and validation of the meteorological model readers can refer to Cassola et al. (2015, 2016).

Generation and propagation of sea waves have been modeled using WavewatchIII®, version 3.14 (Tolman, 2009). A $336 \times$

$180$ regular grid covers the whole Mediterranean Sea with a resolution of $0.1273 \times 0.09$ degrees, corresponding to about $10 km$ at the latitude of $45°$N. Spectral resolution is characterized by 24 bins in direction and 25 frequencies ranging from 0.06 to

$0.7Hz$ with a step factor of 1.1. The output has been recorded hourly in all points of the computation grid for integrated quantities (i.e. significant wave height $H_s$, mean period $T_{m-1,0}$, peak period $T_p$, mean direction $\theta_m$, peak direction $\theta_p$, directional spreading $\Delta\theta$). The validation of the wave hindcast has been carried out through extensive comparison of simulated quantities and wave buoy data (cfr. Mentaschi et al., 2013a, b, 2015) and has already been employed for different applications such as wave energy resource assessment (Besio et al., 2016) and extreme and wave climate analysis (Sartini et al., 2015a, b).

## 3.2 Buoy data

The Italian Sea Wave Measurement Network (Rete Ondametrica Nazionale RON) started operating in July 1989 (De Boni et al., 1992; Arena et al., 2001; Corsini et al., 2004). The locations of the buoys are indicated in Fig. 2. Until 1998 the network was made by eight pitch-roll directional buoys located offshore, in deep water conditions, of several sea areas equally spaced along the italian peninsula. These original eight stations were: La Spezia, Alghero, Ortona, Ponza, Monopoli, Crotone, Catania and Mazara del Vallo. The statistical wave parameters (i.e. significant wave height $H_s$, mean period $T_m$, peak period $T_p$, mean direction $\theta_m$) were originally retrieved every three-hours, below a station-dependent threshold for $H_s$, and every half an hour above this threshold. The wave data time series, measured by the RON buoys, that have been analysed in the present study, cover a time window of 20 years, from the summer of 1989 until the spring of 2008 for the original eight buoys. For the cluster analysis performed using the RON records, data every three hours were considered for all the stations.

## 4 Results

### 4.1 Comparison between hindcast and buoy measurements

In order to assess the reliability of the hindcast time series related to storm cluster analysis, the results of AF for the RON buoys are analysed and compared to the corresponding grid points of the hindcast model. **These results are shown in figures 5-6**. Results obtained on the basis of the RON data and hindcast series show a good qualitative and quantitative agreement especially for lower threshold conditions (98% percentile) while for higher threshold (99.5% percentile) tend to present stronger differences, e.g. in Alghero (see figure 5). These findings can be explained by the fact that increasing the threshold limit would select just the most energetic wave conditions that are the most difficult to be reproduced by numerical models (a.o. Cavaleri, 2009) and sometimes to be recorded by wave buoys (breakdown, damages or even loss of the instrumentation). Differences are usually larger for smaller time scales, i.e. $0.5 < \tau < 50$ days and the exponent $\alpha$ attains values around $0.15 - 0.3$. For $\tau > 50$ days the variability is lower and all the curves show slopes of similar value, with $\alpha \simeq 1.1 - 1.2$ on average. **Note that in figures 5-6 and in all the figures showing AF curves, the slopes $\alpha = 0.2$ and $\alpha = 1.15$ are indicated for reference.** These results confirm that the hindcast data and the wave buoys show very similar clustering properties.

In all cases, and consistently between model and data estimates, the AF is greater than one for $\tau$ greater than 12-24 hours (0.5-1 days) and two distinct slopes $\alpha$ are recognizable. The first slope is observed generally between 0.5 to 20-50 days in most of the stations. The second, steeper, slope is observed generally between 50 and 100 days. Transition between the two slopes

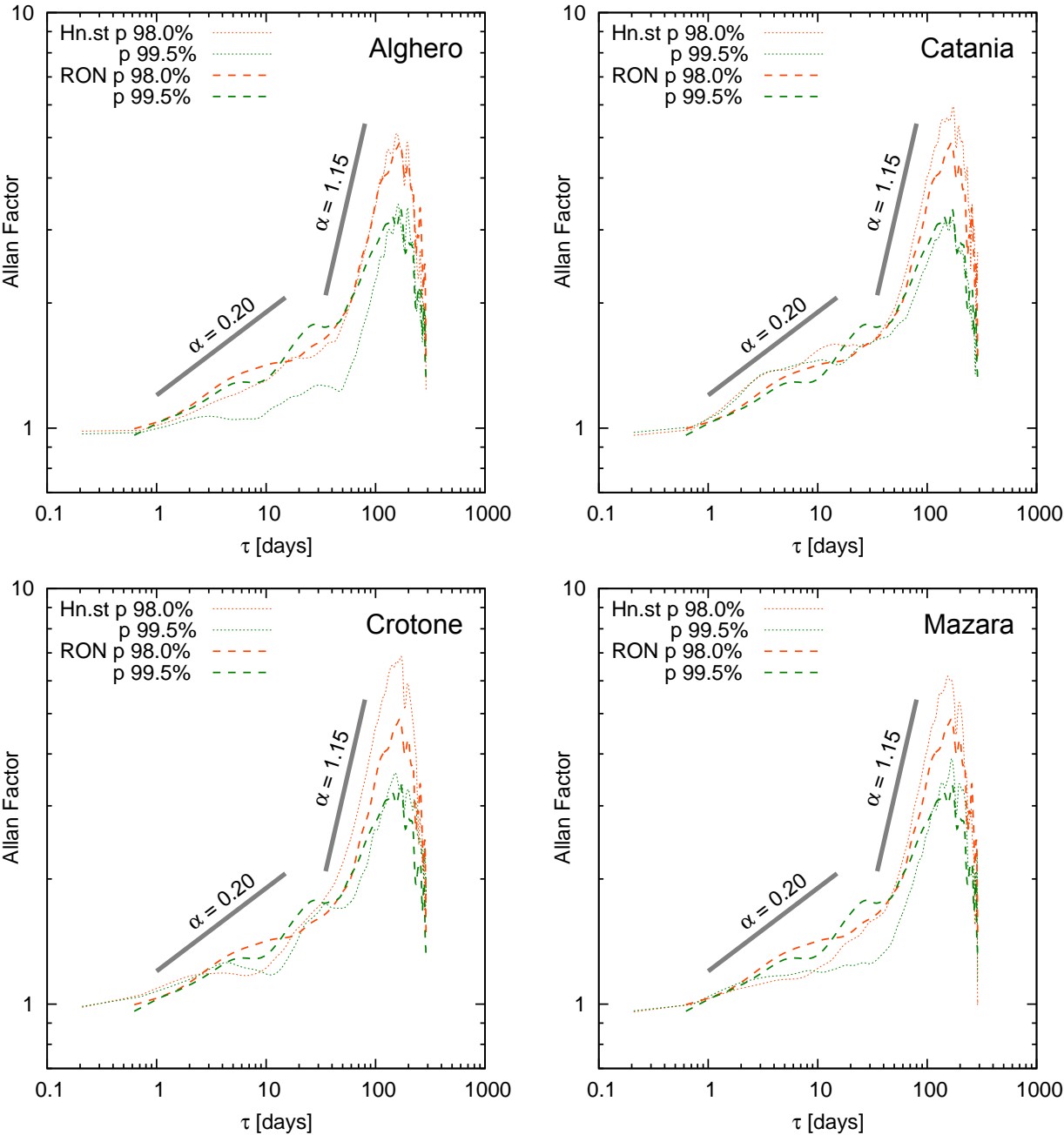

**Figure 5.** Comparison of Allan Factor between RON and Hindcast data series for different threshold percentiles (98% and 99.5%)

differs from station to station. In most stations there is a sharp discontinuity between the two slopes (e.g. Alghero, Ancona, Catania in figure 5). In some cases there is a plateau or a slight decrease in AF value at this discontinuity (e.g. Ponza in figure

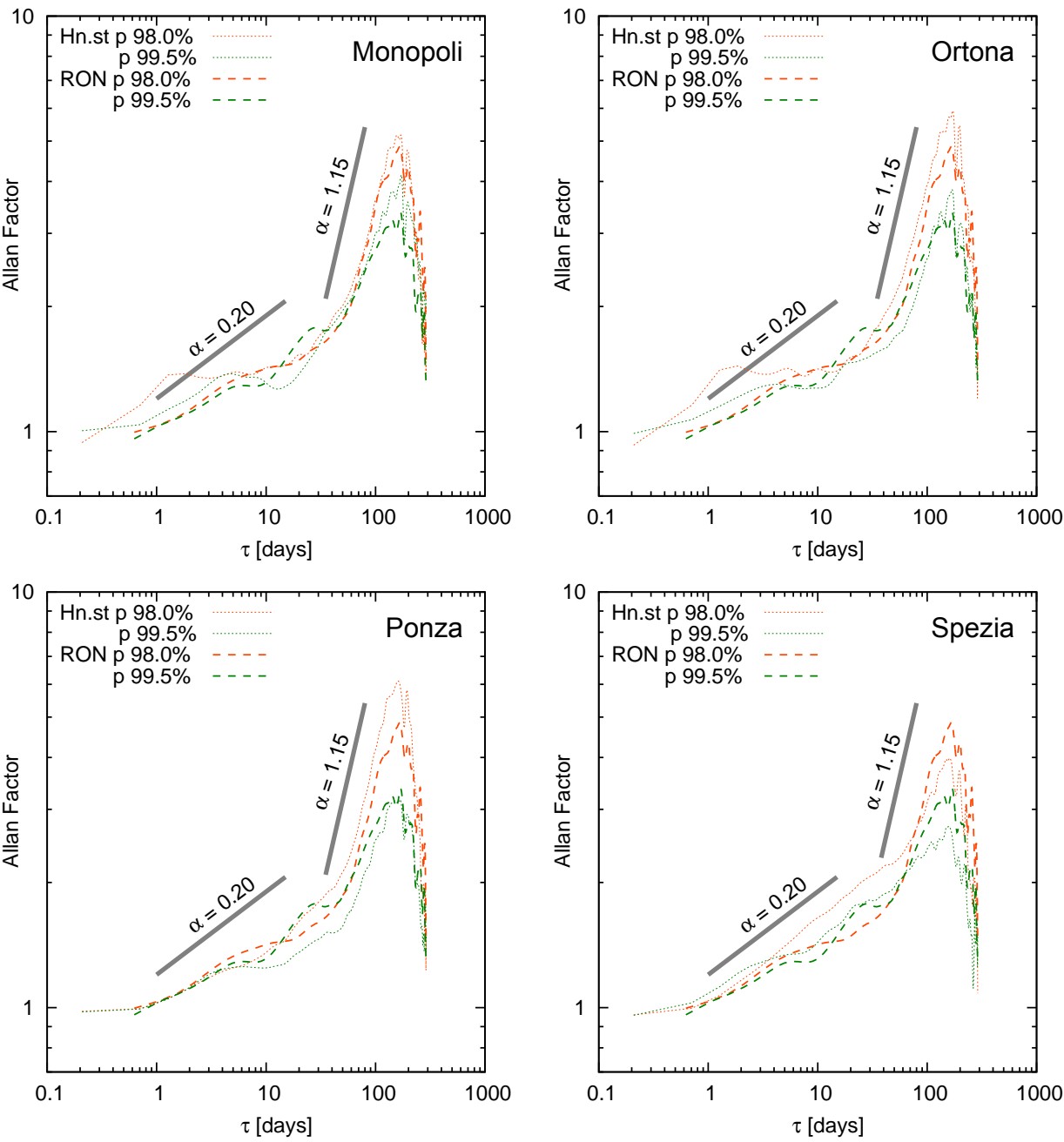

**Figure 6.** Comparison of Allan Factor between RON and Hindcast data series for different threshold percentiles (98% and 99.5%)

6). In few cases a more gentle transition is seen (e.g. in Ortona, see figure 6) that occurs between 10 and 50 days. **In all stations**

**the AF has a maximum at 180 days and it decreases rapidly after, indicating the influence of the annual cycle in the process**.

## 4.2 Comparison with a simulated non-homogeneous point-process

The AF pattern found from data and hindcast time series is compared with that of a simulated non-homogeneous Poisson process. This is generated using the IF technique employed in Serinaldi and Kilsby (2013). The rate function of the simulated hon-homogeneous Poisson process is generated as a sum of sinusoidal components with amplitudes, periods and phases obtained from the Fourier analysis of the reference signal. A Monte Carlo simulation of 1000 time series is then carried out and the simulated population of AF is compared with the reference one. Hindcast points A, G and O (see figure 2) are chosen for this analysis because they show different AF patterns in the time scales $\tau < 50$ days. This analysis reveals that, as expected, the dominant cyclic component for all the considered time series is the one with 1-year period. This was also noted for the RON data in Briganti and Beltrami (2008), where the amplitude of the annual cycle component was estimated to be around $0.25$ m in Alghero, which is consistent with what found in the present work. Together with the annual cycle also the components with periods of six, three, one months and one week have been considered to simulate the non-homogeneous Poisson processes. The results of the comparison are shown in figure 7. For all three points it is clear that the simulated cyclic Poisson process well explains the pattern of the AF at $\tau > 50$ days in all cases. As expected, this is the signature of the annual cycle, which strongly influences the occurrence of above-threshold events. The AF departs from the Poisson distribution at $\tau < 50$ days, above all in points A and G. Note that these results show also that the $\alpha$ for $\tau > 50$ days is always above $1$ and shows very little variability among points. For scales in which a departure from a poissonian behaviour is seen, it has to be noted that this occurs at very low values of $\alpha$, as for example in point $O$. However, data often show oscillations, above all for $\alpha < 0.1$, and it is not possible to make conclusions about the existence of a clustering regime.

## 4.3 AF results over the Mediterranean Sea

Results from the control points located over the basin (see figure 2) are shown in figures 8-11. The AF curves reveal that the characteristics seen in the RON buoys are more general. In particular, three types of AF curves emerge:

a) the first type shows two distinct slopes with a slight decrease of AF between the two. Although the value of $\tau$ that marks the separation between the two threshold varies in space, it can be seen that the change in slope occurs at around $\tau = 20$ days in most cases. This regime is similar to that seen at Ponza buoy and it is very evident at points A (North Thyrrenian Sea), B (Gulf of Lyon), D (Alboran Sea), and E (Algerian Sea) while is not observed for South-East Mediterranean points. **As shown in the previous section this slope indicates a departure from a cyclic Poisson point process**

b) the second type shows three slopes, one for $\tau < 10$ days, one for $\tau > 50$ days, and a transition slope in between. The slope for small values of $\tau$ varies, with $\alpha$ being higher at point R (Adriatic Sea) than elsewhere, e.g. points C (West

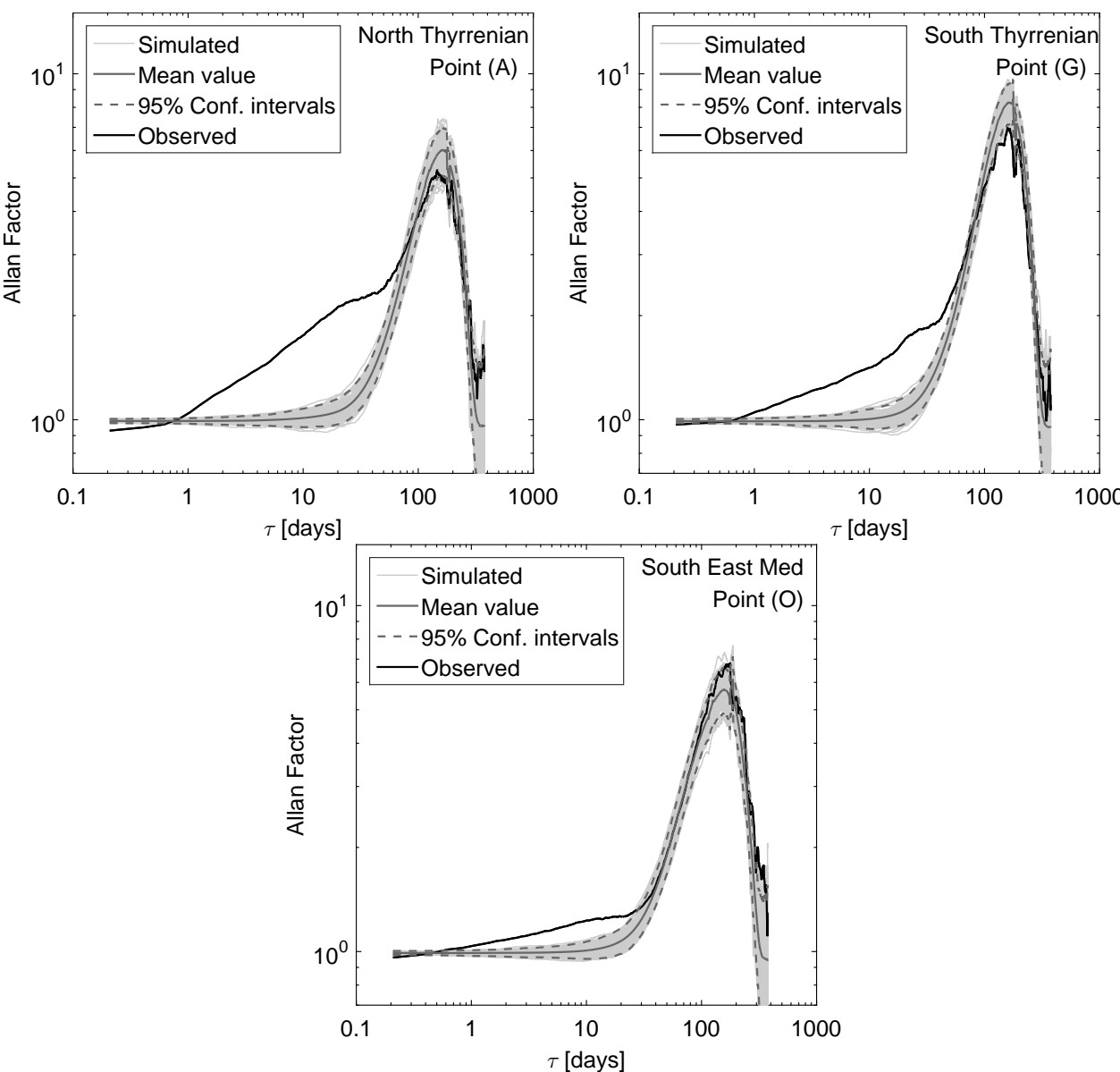

**Figure 7.** Comparison of Allan Factor between Hindcast data series for 98% percentile (black line) and 1000 simulated cyclic Poisson processes (grey lines). The AF corresponding to the 95% percentile of the AF distribution is also plotted (dashed line). Top left, point A (Northern Thyrrenian). Top right Point G (South Thyrrenian). Bottom Point O (South East Med).

Sardinia), F (Tunisian coast), G (South Thyrrenian Sea), M (Ionian Sea), Q (Aegean Sea). At some of these points, e.g. point Q (Aegean), the small time-scales slope approaches $\alpha = 0$.

c) in the third type only one slope is clearly recognizable, generally for $\tau > 20$ days. At smaller scales $\alpha$ can be reasonably considered naught, **indicating that the AF pattern is fully explained by a non-homogeneous Poisson process**. This is the case of the southern Mediterranean points H (Egypt), I (Western Libia), L (North-East Libia), O (South East Mediterranean Sea) and P (Southern Turkey).

A common characteristics of all control points is that $\alpha$ for larger scales shows little variability with the location (i.e. a strong clustering for long period is observed in the whole Mediterranean basin). Also, in all cases as the highest percentile is used $\alpha$ for the small time scales tends to zero.

The spatial distribution of the slopes for the small and large time-scales is shown in figure 12. This figure has been obtained by determining the best fit value of $\alpha$ at different time scales. In order to take into account the local differences in determining the transition between slopes and the different regimes seen in the representative points, the slope for the small time-scales has been estimated using four different ranges of $\tau$. Clustering in the range $12 < \tau < 72$ hours is presented in panel a) while for $12 < \tau < 120$ hours results are showed in panel b). Within this range the small-scale slope is higher in the North-West Mediterranean Sea and, in particular in the North Thyrrenian Sea and in the Balearic Sea. Here $\alpha$ reaches values up to $0.3$. Areas with $\alpha$ around $0.2$ are present in the Adriatic Sea, on the Syrian and Lebanese coast and along the Tunisian coast. The effect of widening the range of $\tau$ is to decrease the best fit value of $\alpha$. This effect reduces the regions that show $\alpha$ significantly higher than zero in particular in the Adriatic Sea and on the East Coast of Tunisia. When the interval $12 < \tau < 240$ hours (0.5-1 days) is used (figure 12 panel c) the best fit of $\alpha$ is significantly higher than zero only in the North-West Mediterranean Sea with the average $\alpha$ around $0.2$ and zones with $\alpha > 0$ are present in the East part of the Adriatic Sea and on the Syrian coast.

The range of $\tau$ for larger time scales is consistent in the whole domain considered when the interval $1200 > \tau > 2400$ hours (50-100 days) is used for best fitting $\alpha$ (figure 12 panel d). Also, the spatial distribution of $\alpha$ for these time-scales is more uniform compared with the previous smaller ranges. $\alpha > 1.0$ is seen in most of the Mediterranean Sea, with peaks in the Levantine Basin, while it becomes lower than unity only in the North Thyrrenian and Adriatic Seas and along Southern Spain and Tunisia. The slope is larger in the Eastern than in the Western sub basins and it is maximum in the Aegean Sea.

## 5   Discussion and conclusions

**The results presented highlighted the presence of two distinct scaling regimes for the arrival of above threshold wave storms: one for time scales shorter than $\tau < 1200$ hours (50 days) that is associated to a departure from the Poisson distribution. This regime is characterised by $\alpha = 0.15 - 0.3$ and is more evident in the North-West of the Mediterranean Sea. In the rest of the basin $\alpha$ is closer to zero and the AF pattern is characterised by oscillations, without a well defined regime.**

**For $\tau > 50$ days the arrival of above-threshold storms is dominated by the effect of seasonal and inter-seasonal oscillations and can be described as a cyclic Poisson process.**. Similar scaling regimes have been observed in other phenomena with seasonal behaviour, e.g. fires (Telesca and Pereira, 2010). It is well known (Lionello et al., 2006, Sartini et al., 2015a) that the Mediterranean Sea is characterised by a very active winter, with the maximum wind speeds found in this season throughout

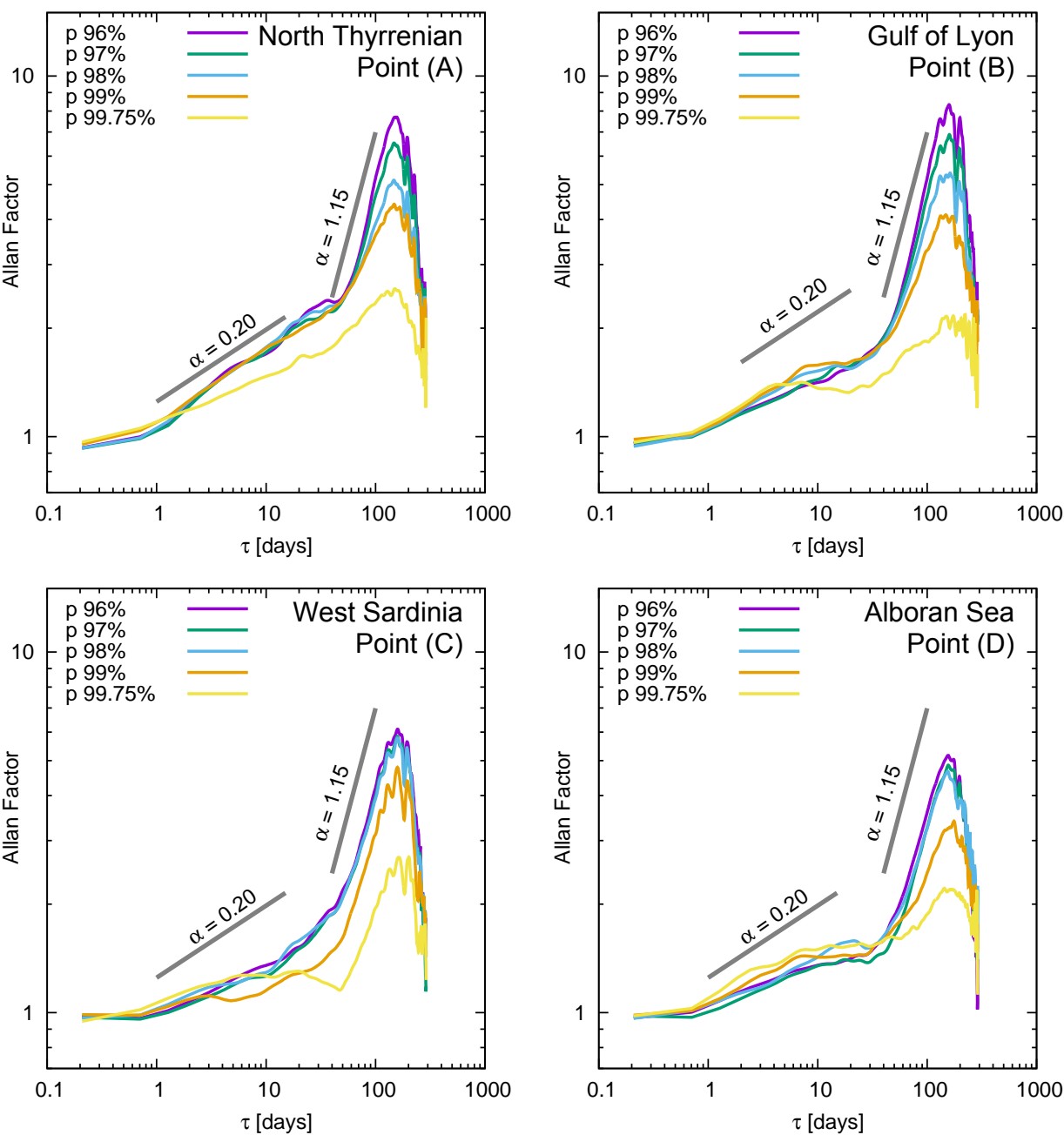

**Figure 8.** Allan Factor (AF) as a function of counting window $\tau$ and of the wave height threshold (different percentiles as in the legend) for different locations in the Mediterranean Sea (cfr. figure 2)

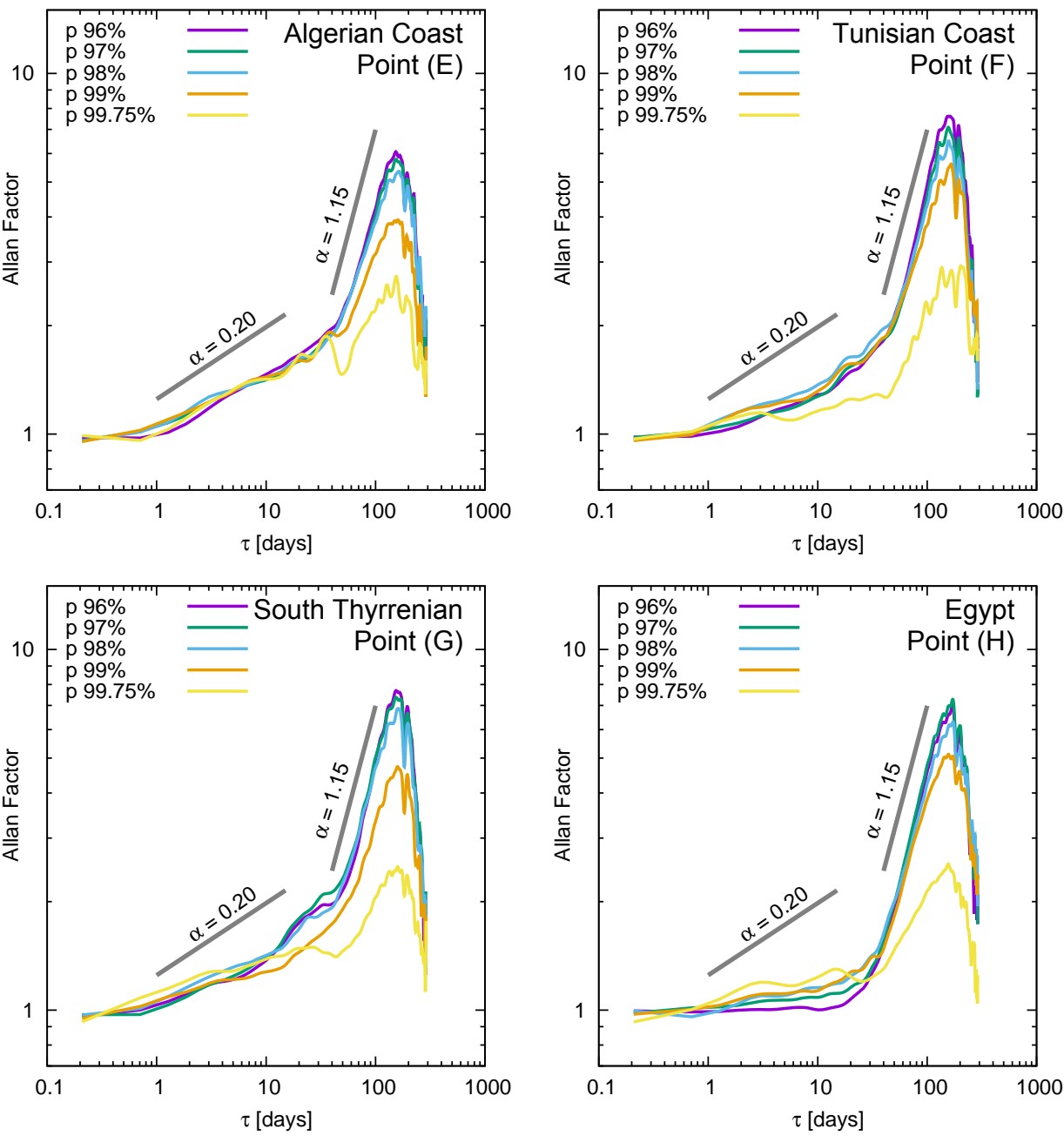

**Figure 9.** Allan Factor (AF) as a function of counting window $\tau$ and of the wave height threshold (different percentiles as in the legend) for different locations in the Mediterranean Sea (cfr. figure 2)

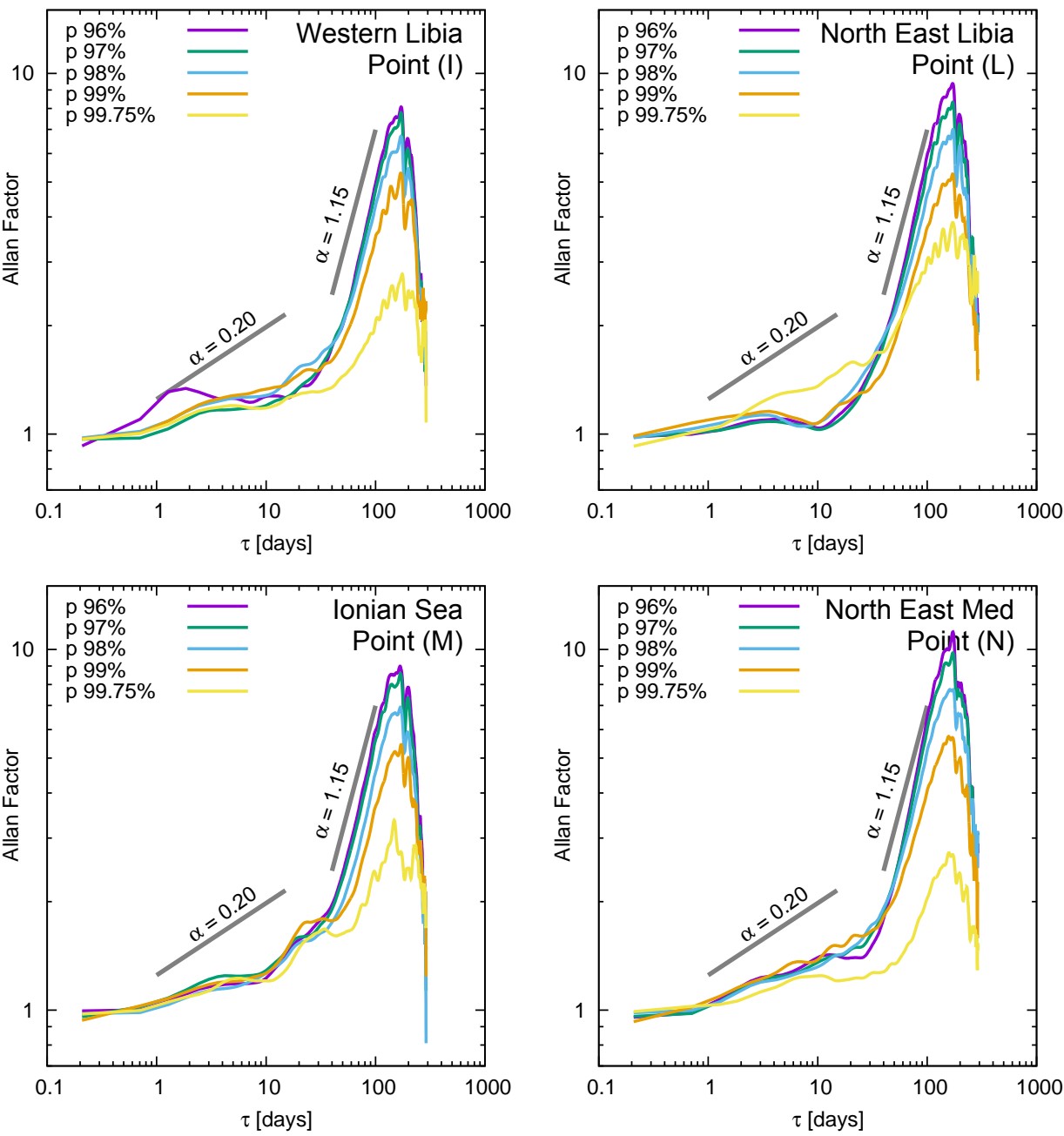

**Figure 10.** Allan Factor (AF) as a function of counting window $\tau$ and of the wave height threshold (different percentiles as in the legend) for different locations in the Mediterranean Sea (cfr. figure 2)

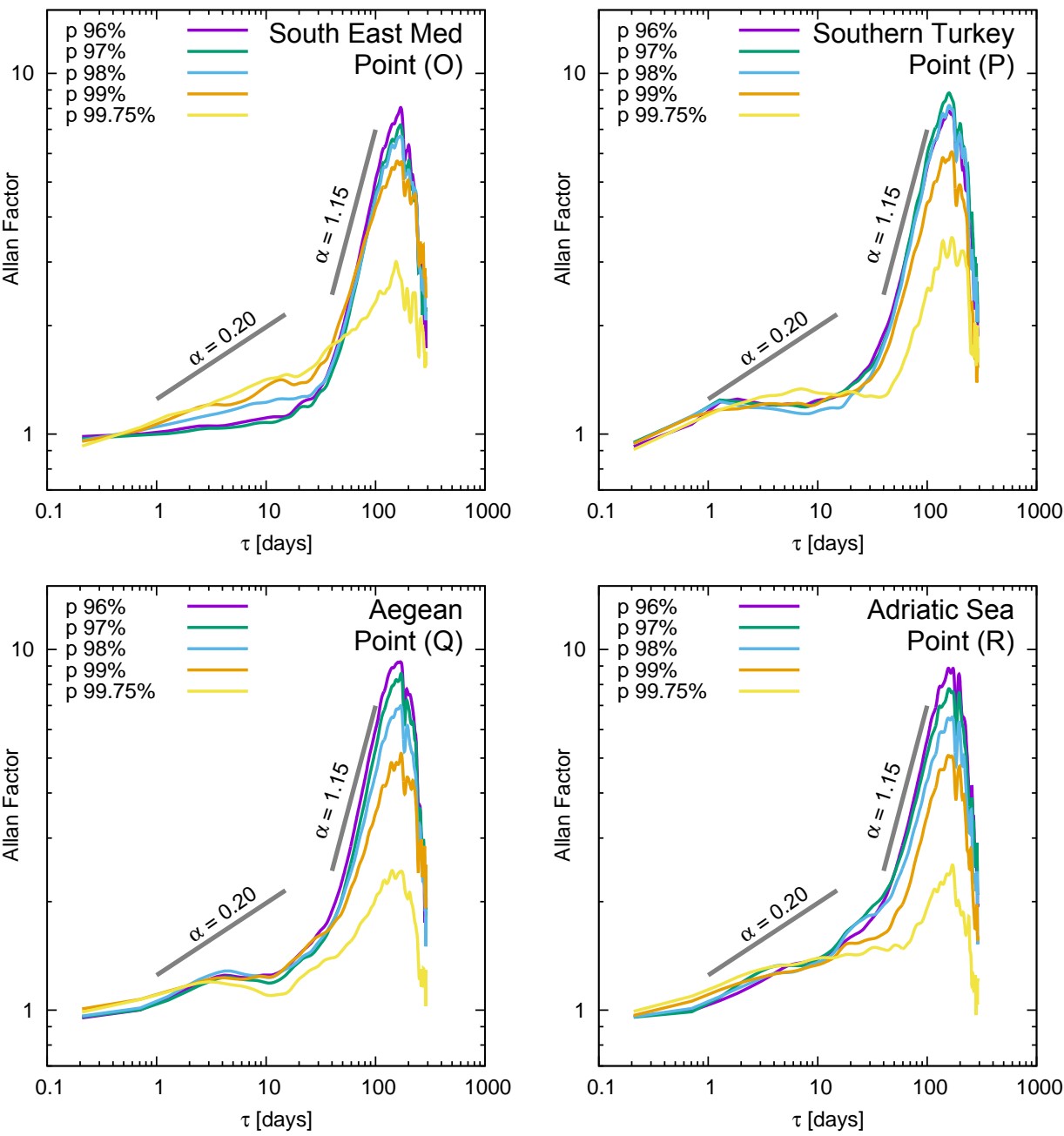

**Figure 11.** Allan Factor (AF) as a function of counting window $\tau$ and of the wave height threshold (different percentiles as in the legend) for different locations in the Mediterranean Sea (cfr. figure 2)

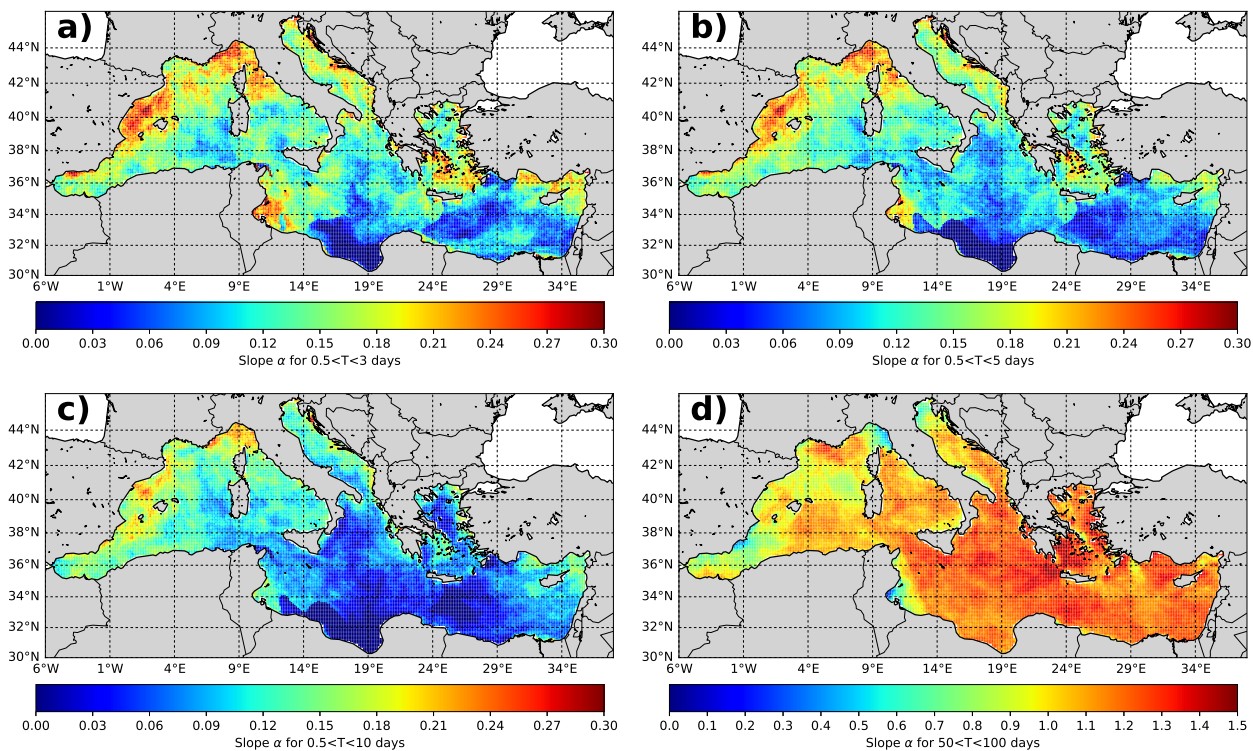

**Figure 12.** Spatial distribution of the exponent $\alpha$ for the whole Mediterranean basin

the basin. On the other hand summer shows the minimum wind intensity and storms are less frequent. Interestingly, although the exponent $\alpha$ **for $\tau > 50$ days is larger than unity in most of the basin, there are local differences, with larger $\alpha$ seen in the South-Eastern Mediterranean, which indicates that the importance of the seasonal components in the arrival of above-threshold events is slightly larger.** These results match with the findings by Sartini et al. (2015a), who found that the southern basin RON buoys (e.g. Crotone, in the Ionian Sea) showed higher seasonality than the Ponza and La Spezia buoy. The latter is located in the Ligurian Sea, a region where $\alpha$ is lower. Although in the region the cyclogenesis in the Gulf of Genoa shows marked seasonality, cyclones are present throughout the year (Lionello et al., 2006, Sartini et al., 2015a) thus reducing the strength of the inter-seasonal clustering. The variability of the AF correlates well with the seasonal cycle of the Mediterranean cyclones as described by Lionello et al. (2016). In this study the Mediterranean is divided in four sectors, (see figure 1 in Lionello et al., 2016). The South East sector, corresponding roughly with the Levantine Basin, is characterised by the highest variability due to the seasonal cycle, while the North West, corresponding to the Northern Thyrrenian Sea, The Gulf of Lyon and Belearic Sea is the most active sector in all seasons. This basin partitioning is revealed by AF results presented in figure 12 panel d).

This persistence of cyclonic events explains also the behaviour at smaller scales (i.e, $\tau < 1200$ hours, 50 days), where the spatial variability is more pronounced. In most of the Southern and Eastern Mediterranean the process is Poissonian at

these scales, while clustering is found in the North-West of the Mediterranean Sea. Here local factors dominate in generating clustering. One example of this is the Gulf of Genoa, where cyclones are particularly frequent and, in average, the pressure is lower than other regions of the Thyrrenian Sea for most of the year (see figure 7 in Sartini et al., 2015a). The clustering at scales of days indicates that meterological conditions favour the occurrence of multiple events in few days. It is not a case that this

behaviour is seen in the most active cyclonic region of the Mediterranean Sea, e.g. the North West according to Lionello et al., 2016. Similar considerations apply to the North Adriatic Sea.

**The values of $\alpha$ found in the present study do not allow to draw conclusions on whether this deviation from a Poisson distribution is large or small for the phenomenon at hand, as there is no comparison with other basins. Because of this, it is important to analyse further basins.** Clustering at these small time scales have the potential to exacerbate local beach

erosion generated by individual storms, as shown in Dissanayake et al. (2015), hence it will be important to understand the implication of the found time regimes on the dynamics of the Mediterranean coastal regions.

*Author contributions.* G. Besio and L. Mentaschi developed the wave hindcast and the Allan Factor analysis for the Mediterranean Sea; R. Briganti coordinated the work and gave the theoretical ideas to develop the analysis; A. Romano and P. De Girolamo developed the analysis for the RON buoy dataset and carried out the comparison with the simulated non-homogeneous Poisson point process. All the authors

participated actively in the preparation and writing of the manuscript.

*Acknowledgements.* The work described in this publication was supported by the European Community's Horizon 2020 Research and Innovation Programme through the grant to HYDRALAB-PLUS, Contract no. 654110. R. Briganti expresses his gratitude to the Engineering and Physical Sciences Research Council (EPSRC) for providing the funding through the FloodMEMORY project (Grant number: EP/K013513/1). **The authors would like to thank Dr Wahl and an anonymous reviewer for having contributed to the improvement of**

**the manuscript. The authors are grateful to Dr Francesco Serinaldi for the proficous discussion during the revision of the manuscript and for having made available the routines for the simulation of cyclic Poisson processes.**

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
