# Peer review of "Time-clustering of wave storms in the Mediterranean Sea"

_Natural Hazards and Earth System Sciences, 2016_

## Short Comment (SC1) · 3 Nov 2016

F. Serinaldi

francesco.serinaldi@ncl.ac.uk

Dear Riccardo and coauthors,

I'm writing only to mention an aspect that can potentially make your results misleading. From Figures 2 and 3, and discussion in the text, I see that the waves show a clear seasonal pattern (as expected); however, from the text, it seems that you did not pre-process the data to remove it. If it is so, please consider that what you see in the AF diagrams can easily be an artefact related to seasonality. In fact, the spike at about 180 days is typical of seasonal time series (seasonality affects all scales below one year). For a comparison and further details on the impact of seasonality on AF diagrams, please have a look at Fig. 4 in

Serinaldi F, Kilsby CG. On the sampling distribution of Allan factor estimator for a ho-

mogeneous Poisson process and its use to test inhomogeneities at multiple scales. Physica A: Statistical Mechanics and its Applications 2013, 392(5), 1080-1089.

Thus, if you did not remove the seasonal pattern, you can easily confuse (apparent) scaling with (true) seasonal fluctuations, which in turn can be explained by non-homogeneous Poisson processes rather than fractal point processes with crossover. As shown in the paper above, in these cases, the AF patterns should be compared with those corresponding to non-homogeneous (seasonal) Poisson processes (to be used as non-trivial benchmark processes); alternatively, you can try to remove the seasonality by data stratification (i.e., making analyses on a seasonal basis).

You can also be interested to

Serinaldi F. On the relationship between the index of dispersion and Allan factor and their power for testing the Poisson assumption. Stochastic Environmental Research and Risk Assessment 2013, 27(7), 1773-1782.

If you need some feedback, please feel free to write to me.

Sincerely,

Francesco

---

## Referee Comment (RC1) · Anonymous Referee #1 · 7 Nov 2016

1) Pag. 5, line 4: delete "of below" 2) Considering the buoy data, the authors analyse three different datasets on the base of their total duration: 20 year, 10 years and 5 years. The total duration of the dataset constrains the maximum available timescale that is respectively about 2 years, 1 year and 6 months. The AF computed for timescales above such maximum does not have so much sense. Figs. 5-7 show the AF for all the data in a time scale range between 0.1 days and about 300 days, which is only consistent with the 10 years long datasets. It would be probably better to show the AF taking into account the reliable maximum timescale for each group of datasets. 3) As rightly observed in the short comment by Serinaldi, the seasonality would affect the AF curve producing that "hump" centered at about 180 days. And this implies that the increase of the AF on the left-side of such "hump" is not a signature of fractal behavior. The suggestion to compare the AF curves of the original data with

those obtained by a cyclic Poisson process is good. However, from a visual inspection it seems that before 20-50 days the AF appears well approximated by a straight line, and the interpretation as a signature of fractality or clustering at timescales below 20-50 days seems to be appropriate. 4) So, the authors may consider to focus their study only on the small timescale ranges for all the data, re-plot the figures and re-discuss the results accordingly. 5) As additional analysis, it would be better to show also the 95% confidence limit for the Poissonian surrogates at each of the considered timescale (in the range below 20-50 days) in order to check the significance of the clustering.

---

## Short Comment (SC2) · 7 Dec 2016

The paper presents the results from analysing observed and modelled wave time series in the Mediterranean to identify temporal clustering of extreme events and the spatial distribution of these clusters. The method that has been chosen is based on the Allan Factor, which is commonly used for this purpose, including a wide range of geophysical data sets. The topic that is discussed is an interesting and important one that has gained more and more attention over the last few years. The paper as it stands is well written and easy to follow and results are presented in a concise way. However, there are a number of things that need to be clarified and/or revised before the paper can be published with NHESS. One aspect I believe should be discussed a bit more is the issue of significance, i.e. how large can AF (or alpha) get by chance? This is important to be able to interpret the results. There is for example a paper by Serinaldi and Kilsby

(2013; http://dx.doi.org/10.1016/j.physa.2012.11.015) where this is addressed.

Below is a list of more specific comments: P1, l. 3. 'spanning the period' P1, l. 9. 'longer scales' P1, l. 12. 'the occurrence' P1, l. 17. Another paper has recently been published where storm surge clusters are investigated in much more detail around the UK: http://www.nature.com/articles/sdata2016107 P2, l. 8. The selection of example references is heavily biased toward one author, if AF is such a prominent method there should be other examples where it has been used. P. 3, l. 3. Close bracket after 'heights' P. 3, l. 10. The way I know AF the denominator should be multiplied by two. If it is just a typo it is an easy fix, but if the analysis has been performed this way everything needs to be repeated. P. 3, l. 12. 'depends on' P. 5, l. 30 to P. 6, l. 7. Somewhere here the authors should refer to Fig. 2 where the locations of the wave buoys are displayed. P. 6, l. 7-10. I don't think this is necessary; it has been said already that only the original stations are used so no need to go into detail what the other station time series look like. P. 6, l. 13ff. What about the "wobbles" that exist at all example sites for the 99.5% threshold in the model data at ∼5 and ∼25 days, it seems to be something systematic. Can it be explained? P. 6, l. 17. Delete 'and Mazara', it is not included in Fig. 5. P. 6, l. 19. 'sometimes' P. 6, l. 21. How can I see from the figures that alpha is between 0.2-0.3 or 1.1-1.2? There is no reference as can be found in the other figures. P. 6, l. 22. 'on average' P. 11-14. Why exactly are the alpha values of 0.25 and 1.15 shown as reference? This is related to my comment above on significance of the results. The same applies to Fig. 12, is it possible to highlight grid points where the results (in this case the slope) is significantly different from zero (or the Poisson assumption)? P. 15, l. 1. Delete 'the' before Southern Spain P. 16, l. 5-7. I didn't understand the last part of this sentence. P. 16, l. 17. 'to exacerbate'

---

## Short Comment (SC3) · 8 Dec 2016

After reading the comments of the two Reviewers, I would like to further highlight a point.

Referee 1 states "it seems that before 20-50 days the AF appears well approximated by a straight line, and the interpretation as a signature of fractality or clustering at timescales below 20-50 days seems to be appropriate", while Dr. Wahl writes "What about the "wobbles" that exist at all example sites for the 99.5% threshold in the model data at $\approx 5$ and $\approx 25$ days, it seems to be something systematic".

As far as I know, wave data exhibit multiple (superimposed) cyclic patterns related not only to annual seasonality, but also to semi-diurnal, diurnal, fortnightly, and lunar (29.5 days) cycles, and others. Maritime/Ocean engineers know this much better than

me. My point is that all these cyclic processes might affect (if not removed) the AF diagram proportionally to their relative amplitude/magnitude within the overall signal. This results in an overlap of "wobbles" whose maximum is centered around the half period of each cyclic component. The evidence of such "wobbles" depends on the relative amplitude (weight) of the cyclic component within the signal. The overlap can also result in apparently local "linear" patterns.

To summarize, before drawing conclusions about scaling/fractality whose attribution might be problematic (especially in a very small range of scales), we need to exclude that the AF patterns result from much simpler (I would say, trivial) well-known properties of the wave signals. Note that AF provides information very similar to other techniques such as classical power spectrum, wavelet spectrum averaged over time, etc., which can therefore provide complementary information.

As far as the significance of the AF patterns is concerned, I stress once again that this should be assessed by comparing the signal with simple but informative (non trivial) point processes. In the present case, a homogeneous Poisson process is not enough (it is trivial), as we already know that the signal is characterized by annual seasonality, at least, and cannot be homogeneous. Actually, a suitable benchmark should be a non-homogeneous point process involving all the main possible cycles characterizing the wave signal. As an alternative (as I mentioned in my previous comment), data should be pre-processed by filtering out such periodic components.

Moreover, in order to support scaling/fractal behavior (or whatever else), simulation of fractal point processes and subsequent comparison of AF patterns are required. Whatever is the explanatory "model" we assume (fractal, cyclic, both, etc.), one needs to show that such a model yields AF patterns (or whatever summary statistic of interest) comparable to the observed ones. In fact, excluding a trivial behavior, such as the homogeneous Poissonian, is not enough to support alternative conclusions on different processes if we do not show that such processes are able to reproduce the observed behavior. This is the rationale of the simulations reported in fig.4 of Serinaldi and Kilsby

(2013), and this seems to me a sound scientific way to test a theory/model/assumption showing its (provisional) validity for operational purposes.

Sincerely,

F

---

## Author Comment (AC1) · 6 Feb 2017

*We thank the anonymous referee for the positive comments on the paper and his suggestions, which we fully taken into account in the revised version of the paper. We modified the manuscript following the reviewers suggestions and highlighted the modified parts in bold in the revised manuscript.*

**1) Pag. 5, line 4: delete "of below"**

*Amended as suggested*

**2) Considering the buoy data, the authors analyse three different datasets on the base of their total duration: 20 year, 10 years and 5 years. The total duration of the dataset constrains the maximum available timescale that is respectively about 2 years, 1 year and 6 months. The AF computed for timescales above such maximum does not have so much sense. Figs. 5-7 show the AF for all the data in a time scale range between 0.1 days and about 300 days, which is only consistent with the 10 years long datasets. It would be probably better to show the AF taking into account the reliable maximum timescale for each group of datasets.**

*We agree on the suggestion by the reviewer about being consistent with different groups of data. We decided to exclude the newer buoys from the analysis. This was done because the group is non-homogenous itself (time series within this group have different lengths) and we had already enough support for the validation from other measurements. Text and figures are changed/removed consequently.*

**3) As rightly observed in the short comment by Serinaldi, the seasonality would affect the AF curve producing that "hump" centred at about 180 days. And this implies that the increase of the AF on the left-side of such "hump" is not a signature of fractal behaviour. The suggestion to compare the AF curves of the original data with those obtained by a cyclic Poisson process is good. However, from a visual inspection it seems that before 20-50 days the AF appears well approximated by a straight line, and the interpretation as a signature of fractality or clustering at timescales below 20- 50 days seems to be appropriate.**

*We agree with the reviewer consideration on the AF patterns and we carried out further analysis to clarify the nature of the processes that occur at different time scales as shown in the AF plots. We did this by using the approach described in Serinaldi and Kilsby (2013). We compared the AF pattern of a time series (hindcast in this case) to the AF distribution of a population of point processes with the same cyclic characteristics and intensity of the reference one. First we used the Fourier analysis to determine the dominant cyclic components in the time series; this led to identify 5 dominant components corresponding to the yearly cycles and, with much smaller amplitudes to cycles with periods of six, three, one months and one week. With the amplitudes and periods of these cycles we simulated a cyclic, hence non-homogeneous Poisson point process. This was done with the Integrate and Fire (IF) technique by Serinaldi and Kilsby (2013), for which Dr Serinaldi kindly provided a script. Subsequently, we compared the distribution of the AF of 1000 realisations of this process to the AF found in the time series under study, results are shown in the new Figure 7. The results of this analysis confirmed the reviewer's conclusions, i.e. that the AF corresponding to time scales longer than 50 days is associated to the cyclic components, while at time scales shorter than 50 days there is a significant departure from a Poissonian AF pattern and, as the reviewer correctly pointed out, there is a clear trend that gives us confidence in interpreting*

*this as clustering. We also included the 95% confidence intervals to further identify the scales at which the AF pattern is within the limits of the cyclic process. We think that this analysis clarifies the significance of alpha and AF.*

*As a consequence of this analysis the paper has been revised and a subsection discussing the results of the comparison with a surrogate population of AF is added and it reads:*

*"4.2 Comparison with a simulated non-homogeneous point-process*

*The AF pattern found from data and hindcast time series is compared with that of a simulated non-homogeneous Poisson process. This is generated using the IF technique employed in Serinaldi and Kilsby (2013). The rate function of the simulated hon-homogeneous Poisson process is generated as a sum of sinusoidal components with amplitudes, 5 periods and phases obtained from the Fourier analysis of the reference signal. A Monte Carlo simulation of 1000 time series is then carried out and the simulated population of AF is compared with the reference one. Hindcast points A, G and O (see figure 2) are chosen for this analysis because they show different AF patterns in the time scales $\tau$ < 50 days. This analysis reveals that, as expected, the dominant cyclic component for all the considered time series is the one with 1-year period. This was also noted for the RON data in Briganti and Beltrami (2008), where the amplitude of the annual cycle component was estimated to be around 0.25 m in Alghero, which is consistent with what found in the present work. Together with the annual cycle also the components with periods of six, three, one months and one week have been considered to simulate the non-homogeneous Poisson processes. The results of the comparison are shown in figure 8. For all three points it is clear that the simulated cyclic Poisson process well explains the pattern of the AF at $\tau$ > 50 days in all cases. As expected, this is the signature of the annual cycle, which strongly influences the occurrence of above-threshold events. The AF departs from the Poisson distribution at $\tau$ < 50 days, above all in points A and G. Note that these results show also that the α for $\tau$ > 50 days is always above 1 and shows very little variability among points. For scales in which a departure from a Poissonian behaviour is seen, it has to be noted that this occurs at very low values of α, as for example in point O. However, data often show oscillations, above all for α < 0.1, and it is not possible to make conclusions about the existence of a clustering regime."*

**4) So, the authors may consider to focus their study only on the small timescale ranges for all the data, re-plot the figures and re-discuss the results accordingly.**

*We clarified the nature of the process at the different scales, hence we retained the AF at those scales and we explained that the process is a non-homogeneous and Poissonian in this region of $\tau$.*

**5) As additional analysis, it would be better to show also the 95% confidence limit for the Poissonian surrogates at each of the considered timescale (in the range below 20-50 days) in order to check the significance of the clustering.**

*As explained above, this has been done and shown in Figure 8.*

---

## Author Comment (AC2) · 6 Feb 2017

**Response to Reviewer 2**

*We thank Dr Wahl for the positive comments on the paper and his suggestions, which we fully taken into account in the revised version of the paper. We modified the manuscript following the reviewers suggestions and highlighted the modified parts in bold in the revised manuscript.*

**One aspect I believe should be discussed a bit more is the issue of significance, i.e. how large can AF (or alpha) get by chance? This is important to be able to interpret the results. There is for example a paper by Serinaldi and Kilsby (2013; http://dx.doi.org/10.1016/j.physa.2012.11.015) where this is addressed.**

*We discussed the nature of the processes described by the Allan Factor (AF) using the same approach described in Serinaldi and Kilsby (2013), according to which the AF pattern of a time series is compared to the AF distribution of a population of point processes that share the same cyclic characteristics and intensity. In order to do so, we followed a series of steps. First we analysed the time series to find the dominant cyclic components; this led to identify 5 dominant components corresponding to the yearly cycles and, with much smaller amplitudes to cycles with periods of six, three, one months and one week. With the amplitudes and periods of these cycles we simulated a cyclic, hence non-homogeneous Poisson point process. Subsequently, we compared the distribution of the AF of 1000 realisations of this process to the AF found in the time series under study. This analysis showed that the AF corresponding to time scales longer than 50 days is associated to the cyclic components, hence the underlying point process is non-homogeneous Poissonian. At time scales shorter than 50 days there is a significant departure from a Poissonian AF pattern. The presence of a clear trend also gives us confidence in regarding this as a non-poissonian process.*

*We think that this analysis clarifies the significance of alpha and AF. Following the approach indicated by Dr Wahl we clarified that alpha=1.15 in the time scales longer than 50 days is consistent with the alpha showed in the cyclic Poissonian population, while alpha=0.15-0.3 at shorter time scales is consistent with a departure from a cyclic Poisson process. As for the significance of alpha, we noticed that although departure from Poissonian is seen at very low values of alpha, it is not possible to distinguish.*

*We modified the text in multiple locations in the paper to reflect these findings and we added a new subsection that describes the tests. You can find the text of the section below and the rest of the modifications in the revised manuscript.*

*"4.2 Comparison with a simulated non-homogeneous point-process*

*The AF pattern found from data and hindcast time series is compared with that of a simulated non-homogeneous Poisson process. This is generated using the IF technique employed in Serinaldi and Kilsby (2013). The rate function of the simulated hon-homogeneous Poisson process is generated as a sum of sinusoidal components with amplitudes, 5 periods and phases obtained from the Fourier analysis of the reference signal. A Monte Carlo simulation of 1000 time series is then carried out and the simulated population of AF is compared with the reference one. Hindcast points A, G and O (see figure 2) are chosen for this analysis because they show different AF patterns in the time scales $\tau < 50$ days. This analysis reveals that, as expected, the dominant cyclic component for all the considered time series is the one with 1-year period. This was also noted for the RON data in Briganti and Beltrami (2008), where the amplitude of the annual cycle component*

*was estimated to be around 0.25 m in Alghero, which is consistent with what found in the present work. Together with the annual cycle also the components with periods of six, three, one months and one week have been considered to simulate the non-homogeneous Poisson processes. The results of the comparison are shown in figure 8. For all three points it is clear that the simulated cyclic Poisson process well explains the pattern of the AF at τ > 50 days in all cases. As expected, this is the signature of the annual cycle, which strongly influences the occurrence of above-threshold events. The AF departs from the Poisson distribution at τ < 50 days, above all in points A and G. Note that these results show also that the α for τ > 50 days is always above 1 and shows very little variability among points. For scales in which a departure from a Poissonian behaviour is seen, it has to be noted that this occurs at very low values of α, as for example in point O. However, data often show oscillations, above all for α < 0.1, and it is not possible to make conclusions about the existence of a clustering regime."*

Below is a list of more specific comments:

**P1, l. 3. 'spanning the period'**

*Amended as requested*

**P1, l. 9. 'longer scales'**

*Amended as requested*

**P1, l. 12. 'the occurrence'**

*Amended as requested*

**P1, l. 17. Another paper has recently been published where storm surge clusters are investigated in much more detail around the UK:**
**http://www.nature.com/articles/sdata2016107**

*We added the reference to the suggested paper.*

**P2, l. 8. The selection of example references is heavily biased toward one author, if AF is such a prominent method there should be other examples where it has been used.**

*We included more references as suggested by the referee These are:*

*Cavers, M. and Vasudevan, K.: Brief Communication: Earthquake sequencing: analysis of time series constructed from the Markov chain model, Nonlinear Processes in Geophysics, 22, 589, 2015.*

*García-Marín, A., Jiménez-Hornero, F., and Ayuso, J.: Applying multifractality and the self-organized criticality theory to describe the temporal rainfall regimes in Andalusia (southern Spain), Hydrological processes, 22, 295–308, 2008.*

*and of course:*

*Serinaldi, F. and Kilsby, C. G.: On the sampling distribution of Allan factor estimator for a homogeneous Poisson process and its use to test inhomogeneities at multiple scales, Physica A: Statistical Mechanics and its Applications, 392, 1080–1089, 2013.*

**P. 3, l. 3. Close bracket after 'heights'**

Amended as requested.

**P. 3, l. 10. The way I know AF the denominator should be multiplied by two. If it is just a typo it is an easy fix, but if the analysis has been performed this way everything needs to be repeated.**

The typo was corrected as indicated. Computations were carried out with the correct formula.

**P. 3, l. 12. 'depends on'**

Amended as requested.

**P. 5, l. 30 to P. 6, l. 7. Somewhere here the authors should refer to Fig. 2 where the locations of the wave buoys are displayed.**

We added the reference to the figure on Page 5 Line 29.

*"The locations of the buoys are indicated in Fig. 2."*

**P. 6, l. 7-10. I don't think this is necessary; it has been said already that only the original stations are used so no need to go into detail what the other station time series look like.**

*We removed the lines as suggested since only the original stations were used.*

**P. 6, l. 13ff. What about the "wobbles" that exist at all example sites for the 99.5% threshold in the model data at ~5 and ~25 days, it seems to be something systematic.**

*We also noticed these oscillations in the AF patterns, however we did not find an obvious explanation of these in comparing our AF pattern to cyclic Poisson Processes. Following the suggestions from Serinaldi we looked at the effect of cycles of period shorter than one week, but they do not influence the AF significantly. In other words we cannot associate the wobbles to any cyclic pattern. Therefore, at the moment, we cannot explain the nature of these oscillations.*

**P. 6, l. 17. Delete 'and Mazara', it is not included in Fig. 5.**

*Deleted as indicated.*

**P. 6, l. 19. 'sometimes'**

*Amended as requested*

**P. 6, l. 21. How can I see from the figures that alpha is between 0.2-0.3 or 1.1-1.2? There is no reference as can be found in the other figures.**

*We amended all the figures and consistently added a trend line for $\alpha$ in all the figures*

**P. 6, l. 22. 'on average'**

*Amended as requested*

**P. 11-14. Why exactly are the alpha values of 0.25 and 1.15 shown as reference? This is related to my comment above on significance of the results. The same applies to Fig. 12, is it possible to highlight grid points where the results (in this case the slope) is significantly different from zero (or the Poisson assumption)?**

*The slopes in the figures are selected as they are representative of the slopes found in the AF curves, hence they are only indicated as reference. We specified this better in the text. We selected alpha=0.2 as representative of the slope for time scales and alpha=1.15 for longer time scales. As for the significance of alpha, as we discussed this previously, it is difficult to identify a well defined regime when alpha<0.1. We clarified this also in the discussion:*

*"The results presented highlighted the presence of two distinct scaling regimes for the arrival of above threshold wave storms: one for time scales shorter than $\tau$ < 1200 hours (50 days) that is associated to a departure from the Poisson distribution. This regime is characterised by $\alpha$ = 0.15−0.3 and is more evident in the North-West of the Mediterranean Sea. In the rest of the basin $\alpha$ is closer to zero and the AF pattern is characterised by oscillations, without a well defined regime."*

*We also concluded that it is not possible to say that the magnitude of the departure from a Poisson process is large or small as, at the moment, there is no comparison with other Seas. We added this consideration in the Discussion and Conclusions section:*

*"The values of $\alpha$ found in the present study do not allow to draw conclusions on whether this deviation from a Poisson distribution is large or small for the phenomenon at hand, as there is no comparison with other basins. Because of this, it is important to analyse further basins. "*

**P. 15, l. 1. Delete 'the' before Southern Spain**

*Deleted as requested*

**P. 16, l. 5-7. I didn't understand the last part of this sentence.**

*This part has been rewritten, following the analysis by Serinaldi and Kilsby (2013).*

**P. 16, l. 17. 'to exacerbate'**

*Amended as requested.*

---

## Author Comment (AC3) · 6 Feb 2017

First of all we would like to thank Dr Serinaldi for his detailed constructive comments on the paper and his subsequent correspondence with us. We are also grateful to Dr Serinaldi for providing a script to carry out the analysis of the AF curves and we acknowledged this in the dedicated section of the paper.

To understand if what we saw in the AF curves was a signature of a departure from a Poisson distribution or that of a non-homogeneous Poisson process we used the approach described in Serinaldi and Kilsby (2013). We compared the AF pattern of a time series (hindcast in this case) to the AF distribution of a population of point processes with the same cyclic characteristics and intensity of the reference one.

Following your suggestions we carried out the following steps:

[Figure]

1- We carried out a Fourier analysis to determine the dominant cyclic components in the time series; this led to identify 5 dominant components corresponding to the yearly cycles and, with much smaller amplitudes to cycles with periods of six, three, one months and one week. Note that cyclic components associated to typical tidal cycles (i.e. 28, 14 days) do not show significant energy. This is expected as the Mediterranean is a microtidal sea. Also the time series analysed are in deep waters, which makes already small sea level oscillations negligible.

2- With the amplitudes and periods of these cycles we simulated a cyclic, hence non-homogeneous Poisson point process. This was done with the Integrate and Fire (IF) technique by Serinaldi and Kilsby (2013). Here we used the script that you kindly provided.

3- Subsequently, we compared the distribution of the AF of 1000 realisations of this process to the AF found in the time series under study. The results of this analysis are shown in Figure 7 of the revised version of the paper.

The results confirm that the AF corresponding to time scales longer than 50 days is associated to the cyclic components, while at time scales shorter than 50 days there is a significant departure from a Poissonian AF pattern. At the moment, however we are unable to draw conclusions on the oscillations that appear at some locations in the scale where departure is detected. We think that further analysis is needed and we will analyse other basins, possibly macro-tidal.

Again we would like to thank you for the very productive discussion during the revision of the manuscript.
* * *

---

## Author Response (AR2)

**Response to Anonymous Reviewer #1**

*We thank the reviewer for his/her further suggestions, which we fully taken into account in the revised version of the paper. We modified the manuscript following the reviewer's suggestions and highlighted the modified parts in bold in the manuscript.*

***"The paper has been substantially revised. However, it still needs a further round of review. The main point is the following. Since the authors have shown that the only significant clustering behavior is observed for tau<50 days, I don't understand why the figures still show the value of alpha for tau>50 days."*** *and* ***"Practically, the author have to better restructure the paper strengthening their results ONLY for tau<50 days, otherwise they risk to convey misleading information."***

*We thank the reviewer for appreciating the further work done for the first revision. We concur with the reviewer that showing the value of alpha for tau> 50 days is misleading as no actual departure from a Poissonian point process occurs at those time scales.*

*Consequently, we eliminated any reference to alpha for tau>50 days from the text and the figures. The reviewer can find the modified sections in bold in the revised manuscript. The affected figures are Figs. 5-6 and Figs from 8 to 12.*

*We also reworked the structure of the paper following the reviewer's suggestion. Specific points raised are explained below.*

***At pag. 6 line 25, the text "For tau > 50 days the variability is lower and all the curves show slopes of similar value, with alpha 1.1−1.2 on average. Note that in figures 5-6 and in all the figures showing AF curves, the slopes alpha = 0.2 and alpha = 1.15 are indicated for reference" has to be rewritten or eliminated because it does not make sense any comment about clustering or alpha slope above 50 days.***

*Eliminated as suggested.*

***The sentence at page 6 line 30 "The second, steeper, slope is observed generally between 50 and 100 days" is misleading because it was shown that above 50 days there is an apparent slope due to the periodicity and not to real intrinsic clustering effects.***

*Eliminated as suggested.*

***Even the discussed transition between two slopes has not to be done, because the only timescale range for which we could discuss about slope is the range below 50 days, because above 50 days there is an apparent slope due to the periodicity.***

*Eliminated as suggested.*

***Probably a better structure of the paper is that first the authors show the results of the AF on the real data, but without any discussion about the slopes, then they perform the cyclic Poisson simulation, and on the base of the comparison derive that the only significant time scale range really related with clustering is that below 50 days.***

*As the reviewer can see we followed his/her suggestion. First we showed the AF data for buoys and model to validate the model AF curves. Then, on the basis of the comparison with a simulated*

*cyclic Poisson process we found the range of time scales at which a departure from a Poisson behaviour is shown and computed the related alpha.*

***Practically, any sentence that refers to the slope for tau>50 days has to be eliminated or re-written, because the real clustering is only below 50 days and not above. For instance, again, at page 16 the sentence "the other hand summer shows the minimum wind intensity and storms are less frequent. Interestingly, although the exponent for tau > 50 days is larger than unity" is misleading because it was shown that for tau>50 the slope is only an effect of the periodicity and does not suggest any intrinsic dynamics related with clustering.***

*As explained before, we eliminated the reference to tau>50 days everywhere. The sentence highlighted here refers to the Discussion and conclusions that have been reworked consequently.*